# Effects of Collagen Hydrolysates on Human Brain Structure and Cognitive Function: A Pilot Clinical Study

**DOI:** 10.3390/nu12010050

**Published:** 2019-12-23

**Authors:** Seiko Koizumi, Naoki Inoue, Fumihito Sugihara, Michiya Igase

**Affiliations:** 1Nitta Gelatin Inc., Osaka 581-0024, Japan; na-inoue@nitta-gelatin.co.jp (N.I.); fumihitosugihara@gmail.com (F.S.); 2Department of Anti-aging Medicine, Ehime University Graduate School of Medicine, Ehime 791-0295, Japan; migase@m.ehime-u.ac.jp

**Keywords:** collagen hydrolysate, cognitive health, magnetic resonance imaging, gray matter volume-brain healthcare quotient, fractional anisotropy-brain healthcare quotient, word list memory test, standard verbal paired-associate learning test

## Abstract

This study investigated the effects of collagen hydrolysates (CH) on language cognitive function and brain structure. In this open-label study, 5 g CH was administered once a day for 4 weeks to 30 healthy participants aged 49–63 years. The primary outcome measures were the brain healthcare quotients based on gray matter volume (GM-BHQ) and fractional anisotropy (FA-BHQ). The secondary outcome measures were changes in scores between week 0 and week 4 for word list memory (WLM) and standard verbal paired associate learning (S-PA) tests as well as changes in the physical, mental, and role/social component summary scores of the Short Form-36(SF-36) quality of life instrument. CH ingestion resulted in significant improvements in FA-BHQ (*p* = 0.0095), a measure of brain structure, as well in scores for the WLM (*p* = 0.0046) and S-PA (*p* = 0.0007) tests, which measure cognitive function. There were moderate correlations between the change in WLM score and the change in GM-BHQ (*r* = 0.4448; Spearman’s rank correlation) and between the change in S-PA score and the change in FA-BHQ (*r* = 0.4645). Daily ingestion of CH changed brain structure and improved language cognitive function.

## 1. Introduction

Dementia is an important health concern. The substantial increase in the proportion of the population aged over 65 years in Japan has been associated with an increase in the prevalence of dementia, including Alzheimer’s disease [1]. The global incidence of dementia was 46.8 million in 2015, and it is expected to increase to 74.7 million by 2030 and 131.5 million by 2050 [2]. Therapeutic interventions to target the modifiable risk factors of dementia are critical in the context of this increased prevalence of dementia. One major modifiable risk factor is diabetes mellitus, which is an attractive target for the development of such interventions. At the 2017 International Conference of Alzheimer’s Disease, the following risk factors were identified as modifiable factors in dementia: depression, obesity, diabetes mellitus, decreased social interaction, and low levels of physical activity [3].

It is now widely accepted that the state of the brain is shown by the cytoarchitecture of the gray matter (GM) and white matter (WM). Dendritic expanses and increases in neuronal synapses are signs of good brain health, and it is thought that these result in high plasticity of the synapses in the GM and so are indicators of flexibility in future learning [4,5]. Thus, the volume of the GM reflects the health condition of the brain [6]. WM plasticity is influenced by factors such as changes in fibrous tissue, myelinogenesis, myelin remodeling, the numbers of oligodendrocytes and astrocytes, and vascularization. [7]. The transmission efficiency of the network between brain domains for axonal fraction anisotropy can be measured by diffusion tensor imaging [8].

A review by Biessels et al. [9] concluded that the incidence of any type of dementia was higher in individuals with diabetes than in those without diabetes, with seven of the 10 reviewed studies reporting this outcome. This suggests it may be possible to manage cognitive function by managing diabetes. Intriguingly, collagen hydrolysates (CH) have recently be shown to provide benefits for patients with type 2 diabetes mellitus (T2DM), including the promotion of glucagon—like peptide 1 (GLP-1) secretion and the inhibition of dipeptidyl peptidase IV activity [10,11]. Furthermore, functional studies in humans have demonstrated that, following the oral ingestion of CH, its three bioactive peptides, prolyl-hydroxyproline (Pro-Hyp), hydroxyprolyl-glycine (Hyp-Gly), and glutamyl-hydroxyprolyl-glycine (Glu-Hyp-Gly), were quickly absorbed into the plasma, where they remained for a long period until being excreted into the urine [12,13,14,15,16]. It has been reported that Pro-Hyp concentrations were lower in individuals with diabetes [17], and that Pro-Hyp was a true inhibitor of tripeptides [18]. In addition, ingesting 5 or 10 g of CH daily for 4–12 weeks has been reported to have beneficial effects for the skin and joints [19,20,21].

Several studies have investigated the effects of CH on the brain. It has been reported that the ingestion of CH may help recovery from brain injury by promoting angiogenesis [22], and that CH exerts neuroprotective action by suppressing inflammatory effects [23]. A study of aged mice reported that marine CH promoted learning and memory [24]. However, potential beneficial effects on structural changes in the human brain have not previously been reported. It is possible that the ingestion of CH may change brain structure and improve cognitive function, potentially helping patients to recover from brain damage.

In this study, therefore, we investigated the impact of oral CH ingestion on structural changes in the brain, as well as cognitive function, in a clinical pilot study that involved healthy individuals. This is the first clinical study to investigate changes in language cognitive function with CH intake.

## 2. Materials and Methods

### 2.1. Collagen Hydrolysates

CH from porcine gelatin (average molecular weight, 1200 Da; Nitta Gelatin Inc., Osaka, Japan) were used for the investigation. The participants took 5 g CH once daily for 4 weeks. This was ingested orally with any type of drink at any time of the day. The participants recorded their daily CH ingestion in a diary throughout the study period. 

### 2.2. Study Design

This open-label study was conducted between October 20 and November 21, 2016 at the Kyoto University Future Research Center. The study was conducted in accordance with the principles of the World Medical Association’s Declaration of Helsinki. The study design and protocol were reviewed and approved by the Institutional Review Board of the Unit for Advanced Research on Human Minds (Approval Number: 27-P-13). The participants provided their written informed consent prior to participation. 

No control group was used because we considered that the placebo effect would be unlikely to affect the MRI (magnetic resonance imaging) analysis of structural changes in the brain. In addition, this was a pilot study to examine CH intake levels, materials, and participants. The dosage used in this study was that shown by our previous clinical studies to have effects such as moisturizing the skin, improving its elasticity, and reducing joint pain.

The primary outcome measures were changes in brain healthcare quotients for the GM volume (GM-BHQ) and fractional anisotropy (FA-BHQ) between baseline and the end of the regular ingestion of CH for 4 weeks. The secondary outcome measures were changes over the same period in scores for word list memory (WLM) and standard verbal paired associate learning (S-PA) tests, as well as changes in the physical, mental, and role/social component summary scores (PCS, MCS, and RCS, respectively) of the SF-36 tool (described below), as indicators of changes in quality of life.

### 2.3. Participants

The study enrolled 30 healthy participants aged 49–63 years (mean age, 56.1 ± 3.6 years; 26 men and four women), recruited internally at Nitta Gelatin Inc. The participants were unpaid. The following exclusion criteria were applied: ingestion of CH or gelatin during the month prior to enrollment in the study; a history of allergies to gelatin or other foods; a diagnosis of a neurological or mental disorder, such as cerebral infarction or dementia; and the inability to undergo functional magnetic resonance imaging (fMRI) or to receive contrast agents.

### 2.4. Evaluation Methods

#### 2.4.1. MRI Acquisition

MRI scans for the study were acquired at week 0 before the start of the study (between 20 October2016 and 24 October 2016) and again at week 4 (between 18 November 2016 and 20 November 2016), at the Kyoto University Future Research Center.

GM-BHQ and FA-BHQ scores, developed by the ImPACT Program for monitoring brain health, were used as the primary outcome measures. Data for these MRI-based quotients were collected using a 3-T scanner, either a Verio (Siemens Medical Solutions, Erlangen, Germany) or a MAGNETOM Prisma (Siemens, Munich, Germany), with a 32-channel head array coil. In brief, a high-resolution structural image was acquired using a three-dimensional T1-weighted magnetization prepared rapid gradient-echo pulse sequence. The parameters were as follows: repetition time, 1900 ms; echo time, 2.52 ms; inversion time, 900 ms; flip angle, 9°; matrix size, 256 × 256; field of view, 256 mm; and slice thickness, 1 mm. Diffusion tensor imaging data were collected with spin-echo echo-planar imaging using generalized autocalibrating partially parallel acquisitions. The image slices were parallel to the orbitomeatal line. The parameters were as follows: repetition time, 14,100 ms; echo time, 81 ms, flip angle, 90°; matrix size, 114 × 114; field of view, 224 mm; and slice thickness, 2 mm. A baseline image (b = 0 s/mm^2^) and 30 different diffusion orientations with a b value of 1000 s/mm^2^ were acquired.

GM-BHQ and FA-BHQ values were calculated using the T1-weighted, T2-weighted, diffusion tensor, and resting-state fMRI images. GM-BHQ provides an assessment of the GM, which includes a wide variety of types of neuron and represents the plasticity of information processing in the brain. FA-BHQ evaluates nerve fibers in the WM and is considered to represent the efficiency of information transmission in the brain [25]. Functional MRI has rapidly become a vital methodology in basic as well as applied neuroscience research. In clinical practice, it has become an established tool for presurgical functional brain mapping [26]. The fMRI data analysis can be viewed at ITU-T H.861.1 Series H: Audiovisual and Multimedia System.

#### 2.4.2. Word List Memory Test 

Mild cognitive impairment presents as declining cognitive function with no effect on activities of daily living [27]. It is used internationally and has been widely validated for assessing cognitive function. The WLM test used in this study was developed by Millenia corporation as the first Japanese version of the test [28].

The WLM test is based on the 10-word recall test of the psychological test batteries CERAD (Consortium to Establish a Registry for Alzheimer’s Disease) and ADAS-Cog (Alzheimer’s Disease Assessment Scale-cognitive subscale). The objective evaluation of cognitive function is achieved using a specific evaluation protocol and algorithm. First, the examiner said 10 words out loud and the participant repeated these immediately three times. Second, the participant picked up the excluded animal a group of three animals 10 times. Third, without help from the examiner, the participant again repeated the initial 10 words; these were recorded by the examiner.

Following a comprehensive assessment, which included the order of the words and answering speed, a memory performance index value for the participant was calculated based on parameters including his or her sex, age, years of learning, ethnicity, and response to the test using algorithms and databases. The resultant patterns were compared with those of participants of the same age and demographic characteristics for the objective evaluation. Testing at week 0 did not affect the test at 4 weeks because the words in the two tests were completely different.

#### 2.4.3. Standard Verbal Paired Associate Learning Test

The S-PA test was developed by the Japan Advanced Brain Dysfunction Society to assess linguistic memory. It comprises 10 pairs of related or unrelated words that are presented to the participant. The examiner read a combination of 10 semantically related words or 10 irrelevant words with no semantic relation, which were memorized by the subject. Next, the examiner presented the first word, and the subject answered with the pair of the word orally. The number of correct answers was scored using a combination sheet of related/unrelated words, and the subjects were tested at weeks 0 and 4. The same set of 10 pairs was used in three trials and the score on the final trial was taken as the outcome measure. Testing at week 0 did not affect the test at 4 weeks because the presentation was completely different.

#### 2.4.4. Measurement of Quality of Life

We used the second Japanese version of SF-36^®^ (iHope International), which has been demonstrated to be reliable and validated [29]. This includes measures of health-related quality of life that can measure subjective outcomes. The PCS, MCS, and RCS scores in this scale were measured at weeks 0 and 4.

### 2.5. Statistical Analyses

The statistical analyses were performed using the medical statistical software STAT Mate III. Data are presented as the mean ± standard deviation (SD). Intragroup comparisons were conducted using the Wilcoxon signed-rank test, with a significance level of 0.05. Correlations between the changes in GM-BHQ and FA-BHQ scores between weeks 0 and 4 and the changes in language cognitive functional scores were evaluated by Spearman’s rank analysis, using the mean changes in each secondary outcome score. The correlation coefficients were interpreted as follows: <0.2, no correlation; 0.2–0.4, weak correlation; 0.4–0.7, moderate correlation; and 0.7–1.0 strong correlation.

## 3. Results

### 3.1. Overall Outcomes

During the study period, one participant discontinued the study as he was unable to arrive at the hospital because of a personal reason. We were unable to assess the S-PA score for five participants because of a lack of time due to their work commitments; however, they completed all the remaining examinations. Thus, the total number of participants was 29, with 24 undergoing the S-PA test according to the protocol (Table 1). No adverse events were observed during the study period.

### 3.2. Brain Structural Changes

There was a significant improvement in the FA-BHQ score at the end of the study period, compared with the baseline, but no significant change in the GM-BHQ score during the same period (Table 2).

### 3.3. WLM and S-PA Test Scores

There were significant improvements in both WLM and S-PA test scores compared to the baseline (Table 3).

### 3.4. Quality of Life Parameters

There were no significant improvements in any of the quality of life parameters (PCS, MCS, or RCS scores) at the end of the study period compared to the baseline values (Table 4).

### 3.5. Correlations Between Brain Structure Scores and Cognitive and Quality of Life Scores

Table 5 presents the correlations among the primary and secondary outcome measures of the study. The changes in GM-BHQ showed a moderate correlation with the changes in WLM scores, and the changes in FA-BHQ showed a moderate correlation with the changes in S-PA scores (Figure 1). There were no other significant correlations. 

## 4. Discussion

The major findings of this study were that, following the daily ingestion of CH over a four-week period, there were significant changes in the participants’ brain structure, as well as improved cognitive function. The results suggested that FA-BHQ, which reflects brain structure, and WLM and S-PA test scores, which reflect cognitive function, were improved by the ingestion of CH. Furthermore, there was a moderate correlation between the changes in FA-BHQ and the changes in S-PA scores, suggesting that CH ingestion resulted in an increase in FA-BHQ, thereby increasing the S-PA score. However, we cannot explain the pattern of the two brain structure measures, GM-BHQ did not show a significant change but the change in that score correlates with the change in one of the memory tests, and the significant change in the other structure measure (FA-BHQ) correlated highly with change in the other memory test.

Furthermore, the mechanism underlying this beneficial effect is unknown. Syouji et al. [30] reported an increase in the expression of brain-derived neurotrophic factor (BDNF) in the hippocampal formation by CH in vitro, and a significant increase in passive avoidance learning. In another study, the oral administration of oyster hydrolytic peptides in normal mice enhanced their spatial learning and memory capacity, accompanied by the upregulated expression of BDNF and neural cell adhesion molecules [31]. These findings in animal models suggest that the ingestion of CH might contribute to the change of brain structure and improvement in language cognitive function through the upregulation of BDNF expression in the brain.

T2DM is associated with cognitive dysfunction and an increased risk of dementia, and T2DM in all age groups has been shown to be associated with a significant increase in the prevalence of diabetes-related cognitive dysfunction [32]. Even in the absence of T2DM, there is a risk of cognitive decline with decreasing insulin sensitivity in elderly individuals [33]. In a clinical trial in which CH was administered to patients with T2DM, we previously demonstrated reductions in the levels of serum hemoglobin A1c (an indicator of diabetes mellitus), fasting blood glucose, and HOMA-IR, indicating improved insulin resistance [11]. It has been reported that T2DM is associated with an increase in amyloid beta accumulation in the brain, with increased insulin resistance shown to increase amyloid β protein accumulation in neurons [34]. It has also been suggested that diabetes mellitus accelerates cognitive dysfunction via cerebrovascular inflammation and amyloid β protein deposition [35]. Importantly, given that T2DM is a risk factor for cognitive decline and dementia, a GLP-1 analog has been reported to exhibit neuroprotective properties and to be a promising therapeutic agent for Alzheimer’s disease [36]. It has been reported that CH promotes GLP-1 secretion and inhibits dipeptidyl peptidase 4 activity in vitro [10].

The participants in the present study were healthy people, without diabetes or cognitive decline. Because of this, it is difficult to judge from the results whether CH improves cognitive ability through changes to insulin secretion. Further consideration is needed to yield any findings to obtain knowledge from improvement of dementia and diabetes by CH intake. No previous studies have investigated the relationship between changes in S-PA and WLM scores and changes in brain structure; this should be explored in future brain function studies. In addition, there was the potential for evaluation bias in the present study because of unblinded testing. Other limitations of this study were that the participants were limited to healthy individuals, only a small amount of data were collected, and the sample size was low.

Future studies are planned to conduct a placebo-controlled, double-blind clinical trial on the effect of CH and to determine the correlations between S-PA and WLM test scores and brain structure. Additional studies are needed to elucidate the mechanism underlying the beneficial effects of CH on brain health.

## 5. Conclusions

The findings of this study suggested that an intervention involving the regular ingestion of CH may have a positive effect on brain structure and may improve cognitive language ability.

## Figures and Tables

**Figure 1 nutrients-12-00050-f001:**
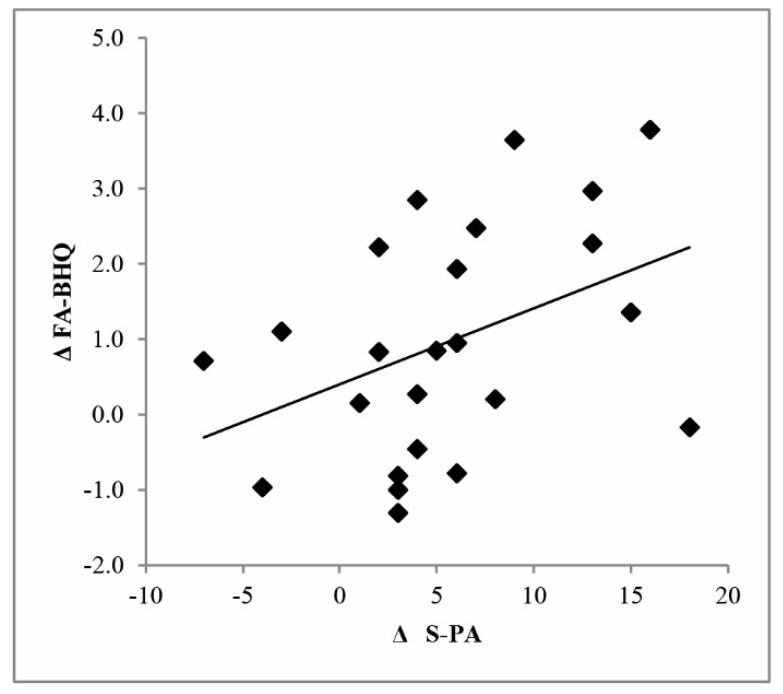
Correlation between the change in brain healthcare quotient based on fractional anisotropy (ΔFA-BHQ) and the change in the standard verbal paired associate learning test scores (ΔS-PA).

**Table 1 nutrients-12-00050-t001:** Mean ages of the participant groups.

	N	Age, Year		
Baseline	30	56.10	±	3.57
Post	29	56.10	±	3.63
S-PA	24	56.08	±	3.82

Data are presented as means ± standard deviation. Post, end of the study period (during which one participant dropped out); S-PA, the participants who underwent the standard verbal paired associate learning test.

**Table 2 nutrients-12-00050-t002:** Changes in the brain healthcare quotient scores based on gray matter volume and fractional anisotropy.

N		Baseline			Post			Δ			*p* (versus Baseline) *
GM-BHQ	29	93.42	±	5.90	93.00	±	5.92	−0.42	±	1.53	0.1415
FA-BHQ	29	94.82	±	4.81	95.73	±	4.46	0.91	±	1.51	0.0095

Data are expressed as means ± standard deviation. ***** Wilcoxon signed-rank test. GM-BHQ, brain healthcare quotient based on gray matter volume; FA-BHQ, brain healthcare quotient based on fractional anisotropy; Post, week 4.

**Table 3 nutrients-12-00050-t003:** Changes in the word list memory and standard verbal paired associate learning scores.

N		Baseline			Post			Δ			*p* (versus Baseline) *
WLM Score	29	67.83	±	6.47	71.06	±	6.39	3.23	±	5.79	0.0046
S-PA Score	24	13.71	±	6.72	19.29	±	6.63	5.58	±	6.18	0.0007

Data are presented as means ± standard deviation. ***** Wilcoxon signed-rank test. WLM, word list memory test; S-PA, standard verbal paired associate learning test.

**Table 4 nutrients-12-00050-t004:** Changes in the quality of life scores.

N		Baseline			Post			Δ			*p* (versus Baseline) *
PCS	29	48.98	±	7.84	51.24	±	7.55	2.25	±	7.44	0.1145
MCS	29	50.17	±	11.43	52.49	±	9.80	2.31	±	6.30	0.0799
RCS	29	50.91	±	12.04	51.35	±	7.35	0.44	±	10.76	0.8457

Data are presented as means ± standard deviation. ***** Wilcoxon signed-rank test. PCS, physical component summary; MCS, mental component summary; RCS, role/social component summary.

**Table 5 nutrients-12-00050-t005:** Correlations among the primary and secondary outcome measures.

N		ΔGM-BHQ		ΔFA-BHQ	
		*r*		*r*	
ΔWLM	29	0.4448	#	−0.0502	
ΔS-PA	24	0.2438		0.4645	#
ΔPCS	29	−0.1340		0.1754	
ΔMCS	29	0.1286		−0.1557	
ΔRCS	29	0.2660		−0.0256	
ΔFA-BHQ	29	0.0567		−	

The data are Spearman’s rank correlation coefficients. # *p* < 0.05. GM-BHQ, brain healthcare quotient based on gray matter volume; FA-BHQ, brain healthcare quotient based on fractional anisotropy; WLM, word list memory test scores; S-PA, standard verbal paired associate learning test scores; PCS, physical component summary; MCS, mental component summary; RCS, role/social component summary.

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
