# Peer review of "Effects of Collagen Hydrolysates on Human Brain Structure and Cognitive Function: A Pilot Clinical Study"

_nutrients, 2019, doi:10.3390/nu12010050_

Round 1

Reviewer 1 Report

In their manuscript, the Authors have investigated the role of collagen hydrolysates (CH) enriched with bioactive oligopeptides on language, cognitive function and brain structure by a clinical study. Despite the interesting topic, the manuscript needs specific improvements.

The Authors declare that the CH from porcine gelatin contained Pro-Hyp, Hyp-Gly, and Glu-Hyp-Gly. How this was evaluated? The relative ratio among these components is known? In literature different doses of CH administration are present. Why did the authors choose 5g die? Result - this section requires a significant improvement. Why the Authors have chosen to start whit the open-label study with respect to placebo-controlled study? I suggest to add a table whit subject characteristic such as age (y), weight (kg), height (cm) and BMI (kg/m2); Conclusion - In this section, the scope of the research should be emphasized.

Author Response

The authors would like to thank for their helpful suggestions and comments. Below, we will respond to each item pointed out by Reviewers.

CH from porcine gelatin contained Pro-Hyp, Hyp-Gly, and Glu-Hyp-Gly

Since the measurement method is not completely established, the ratio cannot be expressed here, so the text was deleted.

Why did the authors choose 5g die? Why the Authors have chosen to start whit the open-label study with respect to placebo-controlled study?

We determined that the placebo effect and assumptions are less likely to affect the MRI structural analysis of brain function in order to verify actual structural changes. This is a pilot study to consider its intake volume, intake material, and target subject etc. We will plan to placebo-controlled study. We added at introduction section of this attached file. (Modified text Line 89-93)

I suggest to add a table whit subject characteristic such as age (y), weight (kg), height (cm) and BMI (kg/m2). In this section, the scope of the research should be emphasized.

There weren't enough people to perform a stratified analysis by age or other status. Therefore, only the average age is shown. (Modified text Line 193)

Conclusion - In this section, the scope of the research should be emphasized.

In this study, whether the collagen peptide affects the structural function of the brain is set as the primary endpoint.

In this study, it was suggested that intervention by ingestion of CH may have a positive effect on brain structure and may improve language recognition. (Modified text Line 300)

We look forward to a good response.

Reviewer 2 Report

Title: Effect of Collagen Hydrolysates on Human Brain 2 Structure and Cognitive Function: A Pilot Clinical 3 Study

Comments:

Abstract- it is important to mention the effect size and/or measures of significance in the results (at least the p- values).

Introduction- It is not customary to use submitted work as a reference unless the journal has provided exclusive authorization with journal name, submission date etc. :” In addition, we 50 suggested that Glu-Hyp-Gly is a neuroprotective biopeptide in the brain (submitted).” Was there any other literature relevant that could support the work? If not, this needs to be worked around differently, but to not add as a citation. It looks incomplete and oftentimes, that lowers the impact of the current work.

“Although the 51 beneficial effects of CH in disorders of the skin and joints as well as in T2DM have been previously 52 investigated, ingesting 5 or 10 g of CH for 4–12 weeks daily is known to be effective on the skin and 53 joints [14-16].” Is there an argument about dosages used for the paper? Or any other justification? If not, this sentence might be better if re-structured.

There must be some literature mentioned to support the peptide connection brain, other than the authors’ previous work to justify the “structural changes in the brain” or “brain damage” connection. T2DM part has been addressed sufficiently.

Materials and methods- “langrage brain function” what does this refer to? Has this been mentioned in the introduction? What’s the connecting link with CH ingestion?

“The studies involving human participants were reviewed and approved by the Institutional Review Board of the Unit for Advanced Research on Human Minds (Approval Number: 27-P-13).” Was there more than one study conducted? Line 67 talks about one open-label study. Please clarify

“In this study, 30 healthy subjects aged between 49 and 53 (average, 56.1 ± 3.6) years participated in the study”

Please see minor similar corrections such as using “this study” twice between lines 81 and 82, throughout the manuscript.

If other studies have published methods for calculating “GM-BHQ and FA-BHQ” as used in the study, please cite appropriately. If this is a novel technique, it is important to provide the details in a supplement, for the purpose of replicability. The techniques as described currently, are unclear “GM-BHQ and FA-BHQ were calculated using the T1-weighted, T2-weighted, diffusion 105 tensor, and resting-state functional MRI images, approved using ITU-T H.861”.

Please also provide appropriate citations for 2.4.2 and 2.4.3 methods that have been validated like 2.4.4.

Please include mean and SD in the statistical analyses section as well as calculation of delta.

Please also describe the main findings from each table instead of presenting them as table. It is important to make relevant connections (descriptive) between the findings and what was hypothesized, in the results section, than simply presenting tables and figures.

This should be a part of the results section. Please also indicate in words, what do these tests signify, and what does that mean in terms of better or worse health “The results of the present study suggested that the FA-BHQ score reflecting changes in brain 187 structure as well as the MCI and S-PA scores reflecting cognitive function were improved by the 188 ingestion of CH. Furthermore, the delta GM-BHQ score moderately correlated with delta MCI score 189 following CH ingestion, indicating that CH ingestion leads to an increase in the delta GM-BHQ, 190 thereby increasing the delta MCI score. The delta FA-BHQ score moderately correlated with delta S-191 PA score following CH ingestion, indicating that CH ingestion leads to an increase in the FA-BHQ, 192 thereby increasing the S-PA score.”

Discussion: This should be the first line for discussion “The major findings of the current study were the change in the brain structure and the 194 improvements cognitive function observed following CH ingestion.”

Is this statement appropriate? “This study indicated that CP ingestion changes brain structure, such as FA-BHQ and GM-BHQ, 227 thereby improving cognitive function, such as MCI and S-PA score.” Does it change brain structure directly? Cause-effect relationships should be carefully explained/interpreted.

Please reword: “Authors should discuss the results and how they can be interpreted in perspective of previous 229 studies and of the working hypotheses. The findings and their implications should be discussed in 230 the broadest context possible. Future research directions may also be highlighted.”

Overall: Very interesting research!

Author Response

The authors would like to thank for their helpful suggestions and comments. Below, we will respond to each item pointed out by Reviewers.

Abstract- it is important to mention the effect size and/or measures of significance in the results (at least the p- values).

Thank you for your advice, we added. (Modified text Line 21,23,24)

Introduction- It is not customary to use submitted work as a reference unless the journal has provided exclusive authorization with journal name, submission date etc.” In addition, we 50 suggested that Glu-Hyp-Gly is a neuroprotective biopeptide in the brain (submitted).” Was there any other literature relevant that could support the work? If not, this needs to be worked around differently, but to not add as a citation. It looks incomplete and oftentimes, that lowers the impact of the current work.

Thank you for your advice, we deleted this sentence. (Modified text Line 59)

“Although the beneficial effects of CH in disorders of the skin and joints as well as in T2DM have been previously investigated, ingesting 5 or 10 g of CH for 4–12 weeks daily is known to be effective on the skin and joints [14-16].” Is there an argument about dosages used for the paper? Or any other justification? If not, this sentence might be better if re-structured.

There is an argument about dosage in those papers, we would like to keep this sentence.

There must be some literature mentioned to support the peptide connection brain, other than the authors’ previous work to justify the “structural changes in the brain” or “brain damage” connection. T2DM part has been addressed sufficiently.

We added reference 19,20 about brain damage. (Modified text Line 60-63)

Materials and methods- “langrage brain function” what does this refer to? Has this been mentioned in the introduction? What’s the connecting link with CH ingestion?

We added reference 21 about cognitive function in vivo by CH ingestion. (Modified text Line 63-65)

“The studies involving human participants were reviewed and approved by the Institutional Review Board of the Unit for Advanced Research on Human Minds (Approval Number: 27-P-13).” Was there more than one study conducted? Line 67 talks about one open-label study. Please clarify

We made a mistake in my expression. Modified to single format. (Modified text Line 86-87)

“In this study, 30 healthy subjects aged between 49 and 63 (average, 56.1 ± 3.6) years participated in the study” Please see minor similar corrections such as using “this study” twice between lines 81 and 82, throughout the manuscript.

We corrected modified text line 102.

If other studies have published methods for calculating “GM-BHQ and FA-BHQ” as used in the study, please cite appropriately. If this is a novel technique, it is important to provide the details in a supplement, for the purpose of replicability. The techniques as described currently, are unclear “GM-BHQ and FA-BHQ were calculated using the T1-weighted, T2-weighted, diffusion 105 tensor, and resting-state functional MRI images, approved using ITU-T H.861”.

Please also provide appropriate citations for 2.4.2 and 2.4.3 methods that have been validated like 2.4.4.

Please include mean and SD in the statistical analyses section as well as calculation of delta.

We corrected modified text line 40-48, 143-151,159-167.

Please also describe the main findings from each table instead of presenting them as table. It is important to make relevant connections (descriptive) between the findings and what was hypothesized, in the results section, than simply presenting tables and figures.

We added the explanation each table and figure.

This should be a part of the results section. Please also indicate in words, what do these tests signify, and what does that mean in terms of better or worse health “The results of the present study suggested that the FA-BHQ score reflecting changes in brain 187 structure as well as the MCI and S-PA scores reflecting cognitive function were improved by the 188 ingestion of CH. Furthermore, the delta GM-BHQ score moderately correlated with delta MCI score 189 following CH ingestion, indicating that CH ingestion leads to an increase in the delta GM-BHQ, 190 thereby increasing the delta MCI score. The delta FA-BHQ score moderately correlated with delta S-191 PA score following CH ingestion, indicating that CH ingestion leads to an increase in the FA-BHQ, 192 thereby increasing the S-PA score.”

We moved those sentences to results section and simplified to consider Reviewer 3 comments.  (Modified text line 232-233)

Is this statement appropriate? “This study indicated that CP ingestion changes brain structure, such as FA-BHQ and GM-BHQ, 227 thereby improving cognitive function, such as MCI and S-PA score.” Does it change brain structure directly? Cause-effect relationships should be carefully explained/interpreted.

This is just a possible outcome and cannot be fully indicated. We corrected the word from “indicated” to “suggested”. (Modified text line 294)

Please reword: “Authors should discuss the results and how they can be interpreted in perspective of previous 229 studies and of the working hypotheses. The findings and their implications should be discussed in 230 the broadest context possible. Future research directions may also be highlighted.”

The last paragraph of the discussion appears to be from an “instructions to authors” document. So, we deleted it. (Modified text line 296-298)

We look forward to a good response.

Reviewer 3 Report

This paper reports a small study examining the impact of dietary supplementation with collagen hydrolysates on the human brain, memory function and quality of life.  Although there is a need for research in this area as this substance is being sold to the public, there are several major issues with this paper.

Firstly, the rationale around T2DM is largely spurious.  The participants in this study do not have T2DM and they are not in the age range in which age related cognitive decline might be expected.  Although the arguments around the possible relevance of collagen hydrolysates to T2DM and cognition in T2DM seem reasonable, they are not relevant to these participants and any effects that might be observed.

Another major issue is the use or description of the memory tests and the interpretation of their results.

I think it is not appropriate to use the label of “MCI’ to refer to the memory test being used in this paper.  MCI is a diagnosis not a test.  In addition, the reference given [20] is not for a Japanese version of any test but a study trying to see if further diagnostic value can be extracted from the list learning task.  The description of the method of this task is not clear.  The CERAD list procedure is for 3 presentation and test trials followed by a delayed test.  There is a description of a ‘comprehensive evaluation” of the recalls to produce an outcome measure but it is not clear what that measure is.

There is no mention of the use of alternate forms of the memory tasks (alternate lists) on the two occasions.  If the same version was used with a 4 week interval some improvement on the second testing would be expected, both on the basis of a general practice effect with the task and some residual memory of the lists.  It appears that the improvement reported could simply be the effect of repeated testing and nothing to do with the intervention.

It is problematic that of the two brain structure measures, one (GM-BHQ) does not show a significant change but the change in that score correlates with the change in one of the memory tests, and the significant change in the other structure measure (FA-BHQ) correlates as highly with change in the other memory test.  This pattern is not readily interpretable.  If the same lists were used on both occasions it means no reliable conclusion can be drawn from this data.

In the first paragraph of page 6 it is incorrectly claimed that there was a change in the GM-BHQ measure.  This paragraph is confusingly worded, by talking about increase in the delta.  Unless I have misunderstood something, delta is the change in scores.  It is simpler to say that the change in one measure was related to the change in the other measure.

On the whole, the brain structure results are interesting but much of the rest of the paper is flawed and does not warrant publication.

Minor comments

Page 1 line 33.  Add “global” before “incidence” and remove  “not only in Japan but also globally”.

Line 44 – should GLP-1 and DPP 4 be given their full names?  It would reduce confusion for less familiar readers.

Page 2 – the suggestion in lines 54-56 is not supported by citations to evidence.  This is a critical claim in justifying the study and either needs to be supported by research or a clearer argument for why an effect might be expected.

Line 81 – “between 49 and 53 (average, 56.1 ± 3.6) years”  This can’t be correct, the mean is outside the range.  This information and the error are repeated on page 4.

Line 82 – “The subjects were recruited at the Nitta Gelatin Inc.”  What does this mean.  Were they employees? Were they paid anything to participate?  How were they made aware of the study?

Page 3 – the S-PA test.  The description of this is a little unclear. On lines 126-127 please make it clear that it is 10 pairs of related or unrelated words that are presented to the participant. Line 130-131 would be better expressed as saying there were 3 presentation/test trials using the same set of 10 pairs and the score on the final trial was taken as the outcome measure.

Line 142 – missing “<” before the first “.7”

Line 148 – not being able to obtain S-PA scores from 5 participants for “personal reasons” is too vague and a better explanation is required.  If they refused to complete the task this should be stated.

Line 221 – “unblinded” not “unblended”

The last paragraph of the discussion appears to be from an “instructions to authors” document.

Author Response

The authors would like to thank for their helpful suggestions and comments. Below, we will respond to each item pointed out by Reviewers.

Firstly, the rationale around T2DM is largely spurious.  The participants in this study do not have T2DM and they are not in the age range in which age related cognitive decline might be expected.  Although the arguments around the possible relevance of collagen hydrolysates to T2DM and cognition in T2DM seem reasonable, they are not relevant to these participants and any effects that might be observed.

This study it is only a hypothesis because it targets healthy people. So, we added explanation in discussion section. (Modified text line 70, 280-283)

I think it is not appropriate to use the label of “MCI’ to refer to the memory test being used in this paper.  MCI is a diagnosis not a test.  In addition, the reference given [20] is not for a Japanese version of any test but a study trying to see if further diagnostic value can be extracted from the list learning task.  The description of the method of this task is not clear.  The CERAD list procedure is for 3 presentation and test trials followed by a delayed test.  There is a description of a ‘comprehensive evaluation” of the recalls to produce an outcome measure but it is not clear what that measure is.

We changed the word from “MCI” to “word list memory (WLM)” and added to 2.4.2 method section. (Modified text line 143-151)

There is no mention of the use of alternate forms of the memory tasks (alternate lists) on the two occasions.  If the same version was used with a 4 week interval some improvement on the second testing would be expected, both on the basis of a general practice effect with the task and some residual memory of the lists.  It appears that the improvement reported could simply be the effect of repeated testing and nothing to do with the intervention.

Word list memory and S-PA were performed using a completely different word, so they never remember the word. We added to method section. (Modified text line 166-167)

It is problematic that of the two brain structure measures, one (GM-BHQ) does not show a significant change but the change in that score correlates with the change in one of the memory tests, and the significant change in the other structure measure (FA-BHQ) correlates as highly with change in the other memory test.  This pattern is not readily interpretable.  If the same lists were used on both occasions it means no reliable conclusion can be drawn from this data.

GM-BHQ may not have a significant difference between before and after ingestion due to large variation. This correlation shows only that the higher the delta GM-BHQ indicated the higher the delta WLM score.

In the first paragraph of page 6 it is incorrectly claimed that there was a change in the GM-BHQ measure.  This paragraph is confusingly worded, by talking about increase in the delta.  Unless I have misunderstood something, delta is the change in scores.  It is simpler to say that the change in one measure was related to the change in the other measure.

Those sentences were moved to results section to consider Reviewer 2 comments and simplified.  (Modified text line 232-233)

On the whole, the brain structure results are interesting but much of the rest of the paper is flawed and does not warrant publication.

Though much remains to be unsettled, we believe we made a few contributions toward the subject.

Thank you very much for your polite points. All subsequent minor points have corrected.

Page 1 line 33.  Add “global” before “incidence” and remove “not only in Japan but also globally”.

Line 44 – should GLP-1 and DPP 4 be given their full names?  It would reduce confusion for less familiar readers.

Page 2 – the suggestion in lines 54-56 is not supported by citations to evidence.  This is a critical claim in justifying the study and either needs to be supported by research or a clearer argument for why an effect might be expected.

Line 81 – “between 49 and 53 (average, 56.1 ± 3.6) years” This can’t be correct, the mean is outside the range.  This information and the error are repeated on page 4.

Line 82 – “The subjects were recruited at the Nitta Gelatin Inc.”  What does this mean.  Were they employees? Were they paid anything to participate?  How were they made aware of the study?

Page 3 – the S-PA test.  The description of this is a little unclear. On lines 126-127 please make it clear that it is 10 pairs of related or unrelated words that are presented to the participant. Line 130-131 would be better expressed as saying there were 3 presentation/test trials using the same set of 10 pairs and the score on the final trial was taken as the outcome measure.

Line 142 – missing “<” before the first “.7”

Line 148 – not being able to obtain S-PA scores from 5 participants for “personal reasons” is too vague and a better explanation is required.  If they refused to complete the task this should be stated.

Line 221 – “unblinded” not “unblended”

The last paragraph of the discussion appears to be from an “instructions to authors” document.

We look forward to a good response.

Round 2

Reviewer 1 Report

The revised version has been improved significantly and authors have done a good job to revised the paper. Now the manuscript is suitable for
publication

Author Response

To Reviewer,

Thank you very much for your agreement.

Sincerely yours,

Koizumi

Reviewer 3 Report

The authors have responded to a number of my concerns and clarified aspects of the method to improve the quality of the paper.  I note that there are a number of places where the standard of English is relatively poor (although the authors have my admiration for working in a foreign language) and this should be corrected.

The argument

GM-BHQ may not have a significant difference between before and after ingestion due to large variation. This correlation shows only that the higher the delta GM-BHQ indicated the higher the delta WLM score.

does not make sense to me.  There is as much variance in the delta values as in the raw scores for the two testing sessions so I do not see how a non-significant difference can produce a difference score that reliably correlates with another variable.  I am not convinced that the memory data show any reliable effects due to the intervention.

Author Response

To Reviewer,

Thank you very much for the review.

We agreed of your opinion about the correlation between GM-BHQ and WML. We deleted the following sentence: (Revised file Line 204-206)
In addition, because there was not enough correction, it corrected again.(Revised file Line 13-14, 246-249)

Also the attached Revised 2 version was used a professional English editing service, fixed extensively, however, the meaning has not changed.

We look forward to a good response.

Best regards,

Koizumi
